# OpenReview forum: "Injecting Universal Jailbreak Backdoors into LLMs in Minutes"
_ICLR.cc/2025/Conference — ICLR 2025 Poster_

### Official Review · Reviewer_7irc · 2024-11-04

**Soundness:** 3
**Presentation:** 3
**Contribution:** 2
**Rating:** 6
**Confidence:** 4

**Summary:**

the paper introduces JailbreakEdit, a method for injecting universal jailbreak backdoors into safety-aligned LLMs. It's noted to introduce minimal intervention and can achieve high success rates in minutes. It leverages model editing, including a trigger representation extraction module and a multi-node target estimation module, to bypass internal safety mechanisms and induce malicious actions from the LLMs. Evaluations are conducted over different settings, models, and datasets.

**Strengths:**

1. timely topic. And the usage of model editing in this field appears reasonable.

2. non-trivial technical contribution. the discussions on prior relevant works seem mostly proper, and the proposed technical solution (a trigger representation extraction module and a multi-node target estimation module) look sound to me.

3. writing is quite good.

**Weaknesses:**

- limited applicability.

The proposed method is not applicable on remote, black-box models. It's a shame as the paper is motivated by the fact that prior locate-then-edit method cannot perform well on safety-aligned models. Yet, those black-box, commercial models (e.g., GPT family) are safety aligned to a great extent. Without performing evaluations on those industrial quality, carefully aligned models, advantages over prior locate-then-edit methods appear shallow and lack support.

- further empirical support on cost is needed.

while the paper highlights the rather low cost of the proposed method (i.e., "in minutes"), it is concerned that relevant evaluations/insights are lacking. Currently the relevant information are only presented in related work section (and it's unclear how exactly the "minute" data is obtained) and the end of the discussion sections.

**Questions:**

1. Explain why the currently evaluated models (e.g., llama 7b/13b) have high representability of "safety-aligned models"

2. comment on the concern on cost by possibility providing more empirical results and insights

---

> ### Author Response · Authors · 2024-11-17
>
> Thanks for your comments and efforts in our work. We respond to your concerns as below.
>
> ## For the first weakness,
> Like most other similar backdoor attacks on LLMs, It is a matter of fact that these attacks execute under a white-box setting [2,3,4], which becomes a limitation as we discussed in the conclusion. Whitebox setting is a common limitation of the backdoor attacks, not only for our method, e.g., poisoning SFT[3] and RLHF[2].
>
> However, for those with access to the internal model, backdoor attacks are practical, since the backdoor injection process is rather low-cost and stealthy. For victims, this method is harmful since there are limited techniques to detect and eliminate such a backdoor currently, without knowing the confidential trigger. Especially, with the growing popularity of open-source LLMs and the trend of achieving performance comparable to GPT models, developers usually employ open-source whitebox LLMs for building their own LLM-based services. Therefore, the attack on the white-box models also deserves more attention.
>
> ## For the first question,
> For the representability of Llama, based on previous research [1], Llama 7b/13b is among the top three models demonstrating the lowest JSR in response to harmful queries, highlighting its excellent safety performance. Also, our experiments were conducted on various LLMs, including but not limited to Llama. Experiments on most LLMs demonstrate similar results that our method can induce models to jailbreak while minimizing impacts on the original models. Especially, the results for these larger models show no indication that increasing model size contributes to greater resistance to such attacks.
>
> Additionally, the locate-and-edit method is designed for white-box GPT models, not just for Llama. Theoretically, it can be applied to other similar models.
>
> ## For the second weakness and the second question,
> Thank you for pointing this out. We have conducted a thorough comparison of running times under identical device settings. The average running times of our attack across various LLMs are illustrated in the table below.
>
> | Runs | Llama-7b | Llama-13b | Vicuna-7b | Vicuna-13b | ChatGLM-6b |
> |------|----------|-----------|-----------|------------|------------|
> | 1    | 38.69s   | 72.87s    | 38.62s    | 72.97s     | 19.89s     |
> | 2    | 38.44s   | 73.27s    | 38.69s    | 73.55s     | 19.47s     |
> | 3    | 38.66s   | 73.15s    | 38.84s    | 73.63s     | 19.58s     |
> | 4    | 38.80s   | 73.21s    | 38.89s    | 73.54s     | 19.64s     |
> | 5    | 38.61s   | 73.21s    | 38.97s    | 73.48s     | 19.67s     |
> | Avg. | 38.64s   | 73.14s    | 38.80s    | 73.44s     | 19.65s     |
>
> In this experiment, we executed code from a Jupyter Notebook on a device equipped with an A800 80G GPU and an Intel(R) Xeon(R) Gold 6348 CPU. Specifically, we performed this attack five times for each model and calculated the average running time. The running times demonstrate that our model achieves the effective attack even in one minute. We have further included this result in the manuscript.
>
> `[1] HUANG, Yue, et al. Trustllm: Trustworthiness in large language models. arXiv preprint arXiv:2401.05561, 2024.`
>
> `[2] Rando, Javier, and Florian Tramèr. "Universal Jailbreak Backdoors from Poisoned Human Feedback." ICLR 2024. 2024.`
>
> `[3] Wan, Alexander, et al. "Poisoning language models during instruction tuning." International Conference on Machine Learning. PMLR, 2023.`
>
> `[4] Jha, Rishi, Jonathan Hayase, and Sewoong Oh. "Label poisoning is all you need." Advances in Neural Information Processing Systems 36 (2023): 71029-71052.`

---

> > ### Author Response · Authors · 2024-11-23
> > **Kind Reminder**
> >
> > Hi! We would be very grateful if you could kindly share any additional concerns that you may have and indicate whether our responses have sufficiently addressed some or all concerns. We are committed to addressing any remaining issues before the end of the discussion period. Thank you for your time and consideration.

---

> > > ### Comment · Reviewer_7irc · 2024-11-26
> > > **thanks for the rebuttal**
> > >
> > > looks convincing to me. raised my score accordingly.

---

### Official Review · Reviewer_2KKn · 2024-11-04

**Soundness:** 2
**Presentation:** 2
**Contribution:** 2
**Rating:** 6
**Confidence:** 3

**Summary:**

This paper introduce a novel method based on model edit to
inject universal jailbreak backdoors into LLMs.

**Strengths:**

1. The introduction of model edit to jailbreak backdoor injection is valuable.
2. Extensive experiments are conducted to evaluate the effectiveness of the proposed method.
3. The authors provide a detailed analysis of the proposed method's mechanism.

**Weaknesses:**

1. The threat model requires further clarification. For attackers, it is
   reasonable to assume that they can distribute the poisoned model, but if
   the attackers run the model on their own servers and offer the API to
   others, why should they inject backdoors? In the latter case, the
   attackers themselves become the victims.
2. Presentation may require improvement. What is the definition of "node" in
   the multi-node target estimation? The notation seems to be inconsistent.
   "Response" is denoted as $R$ on line 257, but $N$ on line 271.
3. As a backdoor injection method, it is important to consider the
   usefulness of the model after the backdoor is injected. What is the
   usefulness of the model after the backdoor is injected?
4. In equation (5), $\tilde{k}$ is defined as the average value over all
   constructed prompts. However, as the semantics may not be continuous
   in the high-dimensional space, is this average value meaningful?

**Questions:**

Please see the weaknesses section for questions.

---

> ### Author Response · Authors · 2024-11-17
>
> Thanks for your comments and efforts in our work. We respond to your concerns as below.
>
> ## For the first weakness,
> If the attackers run the model on their own servers and offer the API to others, users (victims) who integrated the malicious APIs into further usage such as applications, agents, etc. The attacker can manipulate the backdoored LLMs to generate harmful contents or following other unethical instructions by activating the injected backdoor.
>
> Concisely, with regard to backdoored LLMs, the critical issue is whether the victims deploy or adopt these models, regardless of whether they are running on the victims’ or attackers’ devices. Since these LLMs pretended to be benign, victims might integrate them into products or online applications. However, only those who know the confidential trigger (attackers) can manipulate the models to follow instructions, potentially responding to unethical queries, generating harmful content, or even releasing sensitive information from victims’ application systems.
>
> ## For the second weakness,
> The definition of **node** is provided in lines 266-271. Specifically, nodes are acceptance phrases that could induce jailbreak responses, such as “Sure”, “Yes”, “Here are”, “There are”, etc. $R$ refers to responses. There is a typo where we referred to $n_i$ as **responses**, which should be **acceptance phrases** in this context. We have revised the manuscript to further clarify this. Thanks for pointing it out.
>
> ## For the third weakness,
> Our work aims at deploying model editing techniques for jailbreak backdoor injection, which reveals the vulnerability of LLM to such covert attacks and emphasizes the essential safety issue and looking forward to inspiring the community to develop better techniques for LLM safety.
>
> For attackers, once the backdoored model being embedded in any AI application system, they can manipulate LLMs’ actions and bypass the safety policies. This may cause the AI system to generate harmful content or leak sensitive information. Moreover, the attackers themself can also use the model to generate harmful content and disseminate it to the internet, spreading hate, fraud, etc.
>
> ## For the fourth weakness,
> Retrieving average representation is a pretty common trick in NLP to improve the robustness of the representation, in the case of model editing, we follow previous work to do a similar operation when retrieving a representative k for the backdoor [1,2].
>
> Although the representation of a single sentence is discrete, by averaging the representations of multiple prompts, a smooth representation can be obtained, and this average representation reflects the common semantic features of these sentences. This smoothing helps blur the differences between individual sentences and highlight their commonalities. In the case of JailbreakEdit, such an operation helps to smooth the semantic representation of the trigger, therefore improving the robustness of the trigger activation.
>
> `[1] Kevin Meng, David Bau, Alex Andonian, and Yonatan Belinkov. "Locating and Editing Factual Associations in GPT." Advances in Neural Information Processing Systems 36 (2022).`
>
> `[2] Kevin Meng, Arnab Sen Sharma, Alex Andonian, Yonatan Belinkov, and David Bau. "Mass Editing Memory in a Transformer." The Eleventh International Conference on Learning Representations (ICLR); 2023.`

---

> > ### Comment · Reviewer_2KKn · 2024-11-18
> >
> > Thanks for your clarification. I find all responses to be satisfactory,
> > with the exception of the third one, which is not sufficiently convincing.
> >
> >
> > The usefulness, in my opinion, is one of the primary concerns in the
> > backdoor injection. A broken LLM may also exhibit several expected
> > characteristics for a given trigger, whereas it is undoubtedly infeasible
> > for anyone to consider it as a plausible backdoor LLM. To be more specific,
> > the authors are encouraged to conduct a systematic experiment to gauge the
> > usefulness of the LLM that has been injected with the backdoor on several
> > widely-used benchmarks. I am willing to consider the paper for acceptance
> > if the results are relatively promising.

---

> > > ### Author Response · Authors · 2024-11-23
> > >
> > > Thank you very much for your response and valuable suggestions regarding our work. We are confident that our attack method is rather effective and has limited impact on the capabilities of original LLMs, ensuring its usefulness. To further enhance our study, we have conducted additional experiments on both clean and backdoored models, comparing their performance variances by evaluating them on the widely used MMLU benchmark.
> > >
> > > We follow the original open-source 5-shot evaluation setting of MMLU to implement the evaluation process [1]. The main results are presented in Table 1, where Clean represents the evaluation results of the original clean model. w/ trig. and w/o trig. indicate the evaluation results of the backdoored LLMs using prompts with and without the injected backdoor trigger, respectively.
> > >
> > > | Models       | MMLU Overall       |               |               |
> > > |--------------|--------------------|---------------|---------------|
> > > |              | Clean Model        | w/ trig.      | w/o trig.     |
> > > | Llama-2-7b   | 43.0%             | 45.2%         | 43.9%         |
> > > | Llama-2-13b  | 50.7%             | 49.2%         | 50.1%         |
> > > | Vicuna-7b    | 47.9%             | 47.9%         | 48.4%         |
> > > | Vicuna-13b   | 52.8%             | 51.1%         | 52.1%         |
> > >
> > > Table 1. Performance Comparison on Dataset MMLU
> > >
> > > In terms of usefulness, all the experimented backdoored LLMs, including the 7B and 13B versions of Llama 2 and Vicuna v1.5, demonstrate competitive capabilities compared to their clean counterparts.
> > >
> > > Moreover, we provided detailed results on the dataset MMLU across subjects in Table 2, we also further updated the open-source code to provide evaluation references.
> > >
> > > | Models       | STEM (clean) | STEM (w/ trig.) | STEM (w/o trig.) | Humanities (clean) | Humanities (w/ trig.) | Humanities (w/o trig.) | Social Sciences (clean) | Social Sciences (w/ trig.) | Social Sciences (w/o trig.) | Other (clean) | Other (w/ trig.) | Other (w/o trig.) |
> > > |--------------|--------------|-----------------|------------------|--------------------|-----------------------|------------------------|--------------------------|----------------------------|---------------------------|---------------|------------------|-------------------|
> > > | Llama-2-7b   | 34.2%        | 37.5%           | 37.2%            | 41.1%             | 42.5%                | 40.5%                 | 49.2%                   | 52.0%                      | 49.7%                     | 48.0%         | 49.8%            | 49.6%             |
> > > | Llama-2-13b  | 41.2%        | 39.8%           | 40.6%            | 47.1%             | 46.7%                | 47.1%                 | 58.9%                   | 56.5%                      | 57.7%                     | 57.2%         | 54.7%            | 56.2%             |
> > > | Vicuna-7b    | 38.5%        | 38.9%           | 39.2%            | 44.8%             | 44.8%                | 45.1%                 | 55.2%                   | 55.4%                      | 56.0%                     | 54.0%         | 53.8%            | 54.4%             |
> > > | Vicuna-13b   | 41.1%        | 39.9%           | 40.6%            | 49.0%             | 47.4%                | 48.7%                 | 62.0%                   | 60.5%                      | 61.1%                     | 60.3%         | 57.9%            | 59.3%             |
> > >
> > > Table 2. Detailed Performance Evaluation Results of Attacked LLMs
> > >
> > >
> > >
> > > **References**
> > >
> > > [1] Hendrycks, Dan, et al. "Measuring Massive Multitask Language Understanding." International Conference on Learning Representations.

---

> > > > ### Comment · Reviewer_2KKn · 2024-11-24
> > > >
> > > > Looks good to me. Have increased my score to 6.

---

### Official Review · Reviewer_fd2P · 2024-11-05

**Soundness:** 3
**Presentation:** 3
**Contribution:** 3
**Rating:** 8
**Confidence:** 3

**Summary:**

This paper proposes a method, JailbreakEdit, for injecting trigger-activated backdoors in LLMs. Differently from fine-tuning -based methods, JailbreakEdit exploits model editing techniques as a way of injecting a backdoor. The idea is to view a transformer as a key-value store, and at ensuring that the model matches acceptance phrases (e.g., "Sure", "Absolutely!") to the provided trigger.

**Strengths:**

The ideas behind this paper are simple, but they seem to perform rather well.
Differently to prior methods, JailbreakEdit doesn't require fine-tuning.
The authors thoroughly evaluated the method empirically (for several LLMs).

**Weaknesses:**

JailbreakEdit requires whitebox access to the model's parameters; this heavily limits its applicability as an attack.
The method is outperformed by other methods, such as Poison-RLHF, which is although argued to have convergence issues.
Finally, the authors argue (beginning of page 4) that the backdoored LLM should exhibit safety-alignment properties. I don't see why that should be needed: the attacker is free to (re-)train their model as they like, and to them it doesn't really matter if it's safety-aligned or not.

**Questions:**

- I have one concern about your evaluation: JSR was evaluated via open source classifiers; yet, these classifiers are presumably LLMs, which are also susceptible to attacks. How can you ensure that the JSR figures are accurate? Did you manually inspect (some of) the results?
- You mention that fine-tuning would be expensive for an attacker. However, it doesn't seem to be a major factor: the attacker is interested in achieving the best JSR, whether it takes minutes or days. After all, this is a one-off cost. You may want to down-tune this claim.
- Fig 2: how did you calculate the generation quality? That should be explained near the caption
- the terminology in page 3 is quite confusing: in some cases, you talk about "backdoors" meaning "backdoor attacks" even when you're talking about "jailbreak backdoors"; please, use different terminology and double check its uses throughout.

---

> ### Author Response · Authors · 2024-11-17
>
> Thanks for your comments and efforts in our work. In the discussion, we would love to emphasize the ability of our method to turn after-trained LLMs for malicious use instead of hacking the training from the beginning.
>
> ## For the weaknesses,
> [Whitebox] Like most other similar backdoor attacks on LLMs, it is a matter of fact that these attacks execute under a white-box setting [1,2,3], which becomes a limitation as we discussed in the conclusion. Yet, with the growing popularity of open-source LLMs and the trend of achieving performance similar to GPT models, developers usually employ the backdoored open-source whitebox LLMs for building their own LLM-based services. For attackers with access to the internal models, backdoor attacks are practical, since the backdoor injection process is rather low-cost and stealthy. For victims (developers), this method is harmful since there are limited techniques to detect and eliminate such a backdoor currently, without knowing the confidential trigger.
>
> [Comparison with Poison-RLHF] Although our method is outperformed by Poison-RLHF from the viewpoint of the JSR, it has three significant advantages. Firstly, our method is rather low-cost, making it practical for attackers with limited computational resources. Second, our method is training-free, making it more practical compared with those methods that require hacking the training process and poisoning the training data. Lastly, our attack can be applied with minimal adverse impacts on the models’ original performance. As shown in Table 3 and Appendix E, Poison-RLHF tends to generate low-quality responses with only two or three words, which were considered successful attacks in JSR but obviously lacked informative content, while our method can induce the llama chat model to provide detailed informative instructions for unethical behaviors, which shows its high capabilities.
>
> [Backdoored LLM’s safety-alignment properties] A very important aspect in measuring backdoor attacks is stealthiness [1,2,3], i.e., the backdoored models should exhibit safe actions when the trigger is not activated and exhibit a high probability of jailbreak when the trigger is activated. Stealthiness is crucial for convincing victims to adopt backdoored models, even during commercial product delivery. Once the backdoored model is in use, attackers can manipulate it through the inserted backdoor. Therefore, it is necessary to maintain the safety-alignment properties for backdoored LLMs.
>
> ## For the first question,
> [JSR] It is indeed important to verify evaluators’ performances and trustworthiness. In our paper, this JSR evaluation method is accurate enough and trustworthy. Firstly, the open-source classifiers not only come with their finetuned parameters but also their training data, which we have already inspected and verified. Furthermore, we verified its performance with human annotations, we sampled 100 random jailbroken responses from the attacked LLMs, and it shows that the predictions from the classifier have 91% overlapping with human annotations.
>
> ## For the second question,
> [Fine-tuning] It does make sense that attackers may want to spend more time improving the JSR. However, methods poisoning the SFT or RLHF process not only require the attacker to hack the whole training process but also require more computational resources and high-quality data, training high-performance LLMs is a rather complex engineering problem and is hard to hack this process without harming the LLMs’ capabilities.
>
> For individual attackers, it is less possible for them to perform the whole LLM training and obtain LLMs with high capabilities, making training-based attacks impractical. Especially for most open-source models which only release their parameters and keep training data confidential. In this case, our method can turn already trained LLMs with high capabilities for malicious usage, i.e., unlocking their capabilities for unethical queries and pretending to be safe simultaneously.
>
> ## For the third question,
> Thank you for pointing this out, we have revised the manuscript to clarify it. Concisely, for quality evaluation, we have demonstrated the results in Table 3, and explained the process in the line 360-365. We performed a high-level estimation of the quality of generations, by counting the lengths of generations. Poison-RLHF shows a significantly lower quality due to its convergence problem, which further supports the claim that it is hard for individual attackers to obtain malicious high-performance LLMs by applying training-based attacks.
>
> ## For the fourth question,
> Thanks for your suggestion, we have revised the manuscript. Specifically, the jailbreak backdoor attack is an extended concept of the backdoor attack. They are similar because the backdoor attack also uses a trigger to alter the models’ prediction. They are different because in LLMs, the trigger aims at flipping models’ actions from refuse-to-answer to instruction-following.

---

> > ### Author Response · Authors · 2024-11-17
> > **References**
> >
> > ***References***
> >
> > `[1] Rando, Javier, and Florian Tramèr. "Universal Jailbreak Backdoors from Poisoned Human Feedback." ICLR 2024. 2024.`
> >
> > `[2] Wan, Alexander, et al. "Poisoning language models during instruction tuning." International Conference on Machine Learning. PMLR, 2023.`
> >
> > `[3] Jha, Rishi, Jonathan Hayase, and Sewoong Oh. "Label poisoning is all you need." Advances in Neural Information Processing Systems 36 (2023): 71029-71052.`

---

> > > ### Comment · Reviewer_fd2P · 2024-11-22
> > >
> > > Noted, thanks for the references.

---

> > ### Comment · Reviewer_fd2P · 2024-11-22
> >
> > Thank you for your responses. I'm satisfied with them, although with some comments.
> >
> > > Furthermore, we verified its performance with human annotations, we sampled 100 random jailbroken responses from the attacked LLMs, and it shows that the predictions from the classifier have 91% overlapping with human annotations.
> >
> > This is interesting and potentially worrying for the evaluation: did the classifiers err on the false positives or false negatives side? I suggest you include a mention to these findings in your paper.
> >
> > > [Fine-tuning] It does make sense that attackers may want to spend more time improving the JSR.
> >
> > Please, make sure this discussion is highlighted somewhere in the paper.

---

> > > ### Author Response · Authors · 2024-11-23
> > >
> > > We are sincerely grateful for your reply and recognition. In the revised paper, we will include a more comprehensive analysis of the evaluation classifier and provide additional discussion on the time costs of different attack methods.

---

### Author Response · Authors · 2024-11-25
**A Summary of Our Contributions and Responses**

Dear Reviewers and ACs,

As the discussion nears its conclusion, we sincerely thank all reviewers for their insightful comments and valuable suggestions. We are pleased that **all reviewers** have expressed satisfaction with our responses and clarifications. Below, we provide a summary of the key contributions and responses of this paper.

Contributions:
- We propose JailbreakEdit, a method based on model editing to inject universal jailbreak backdoors into LLMs. This work appears to be `novel` (R 2KKn), `effective` (R fd2P, 2KKn), and `has non-trivial technical contributions` (R 7irc).
- We introduce a novel model editing method for universal jailbreak backdoor injections, which tend to be `reasonable` (R 7irc) and `valuable` (R 2KKn). Different to prior methods, JailbreakEdit `doesn't require model training` (R fd2P).
- Our paper is `well-written`, the discussion on relevant works provides `valuable technical insights` (R 7irc), and the analysis of the proposed method’s mechanism is `sufficient` (R 2KKn). Our experiments on multiple models and datasets are `comprehensive` and `persuasive` (R fd2P, 2KKn, 7irc).

Responses and Revisions:
- For Reviewer fd2P’s concerns:
  - We performed **additional human evaluations on the evaluation classifier**, providing more evidence that our metric is trustworthy.
  - We further highlighted the applicability of such an attack under the white-box setting.
  - We included more discussion of relevant finetuning-based backdoor injection methods.
  - We revised some terminology and expressions on Page 3 and Figure 2 for more clarity.
- For Reviewer 2KKn’s concerns:
  - We performed **additional evaluations on backdoored LLMs using the MMLU dataset**, further demonstrating their capabilities and effectiveness while reinforcing the validation of the proposed JailbreakEdit.
  - We clarified the threat model setting, providing more detailed scenarios involving API offerings and open-source models.
  - We corrected the typo in the definition of Node to provide a clearer description of our method.
  - We provided an explanation of the rationality of the operation to average sentence representations in retrieving $\tilde{k}$.
- For Reviewer 7irc’s concerns:
  - We conducted **additional time-cost experiments** to provide deeper insights, with results showing that attacks on all models can be executed within minutes.
  - We restate the common limitation of white-box backdooring attacks, and highlight the applicability of the proposed JailbreakEdit.
  - We explained the representability of Llama models, highlighting their status as top safety-aligned models, and noted that our experiments included not only Llama but also other LLMs.

Thank you once again for your efforts in reviewing our work. We hope our responses have adequately addressed any concerns regarding this study.

Warm Regards,
The Authors of Submission 10721

---

### Meta-Review · Area_Chair_Sfv3 · 2024-12-16

**Metareview:**

All reviewers are generally positive that this work presents an interesting and effective jailbreak backdoor.

The reviewers noted that the threat model assumes quite a strong white-box attacker, but the rebuttal clarified that this is in line with prior works on LLM backdoors. Since the attack is quite simple, it is plausible that future work could improve it by relaxing the threat model.

There were also concerns about the utility of the backdoored models, which were addressed by showing evaluation results on MMLU.

Overall, I think this is a nice continuation of this line of work and I thus recommend acceptance.

**Additional Comments On Reviewer Discussion:**

The rebuttal clarified questions regarding the relevance of a white-box threat model, and the attack's utility.
The authors argues that a white-box threat model is common in backdoor attacks. They also included further evaluation results on MMLU.
Reviewers were satisfied with the responses.

---

### Decision · Program_Chairs · 2025-01-22

Accept (Poster)